# Mechanical Rupture-Based Antibacterial and Cell-Compatible ZnO/SiO_2_ Nanowire Structures Formed by Bottom-Up Approaches

**DOI:** 10.3390/mi11060610

**Published:** 2020-06-24

**Authors:** Taisuke Shimada, Takao Yasui, Akihiro Yonese, Takeshi Yanagida, Noritada Kaji, Masaki Kanai, Kazuki Nagashima, Tomoji Kawai, Yoshinobu Baba

**Affiliations:** 1Department of Biomolecular Engineering, Graduate School of Engineering, Nagoya University, Furo-cho, Chikusa-ku, Nagoya 464-8603, Japan; s6a6n6d6o6n9o9m9e7s7h7i77000@yahoo.co.jp; 2Institute of Nano-Life-Systems, Institutes of Innovation for Future Society, Nagoya University, Furo-cho, Chikusa-ku, Nagoya 464-8603, Japan; kaji@cstf.kyushu-u.ac.jp; 3Japan Science and Technology Agency (JST), PRESTO, 4-1-8 Honcho, Kawaguchi, Saitama 332-0012, Japan; kazu-n@g.ecc.u-tokyo.ac.jp; 4Department of Applied Chemistry, School of Engineering, The University of Tokyo, 7-3-1, Hongo, Bunkyo-ku, Tokyo 113-8656, Japan; yanagida@g.ecc.u-tokyo.ac.jp; 5The Institute of Scientific and Industrial Research, Osaka University, 8-1 Mihogaoka, Ibaraki, Osaka 567-0047, Japan; kawai@sanken.osaka-u.ac.jp; 6Institute for Materials Chemistry and Engineering, Kyushu University, 6-1 Kasuga-koen, Kasuga, Fukuoka 816-8580, Japan; masakana91@cm.kyushu-u.ac.jp; 7Department of Applied Chemistry, Graduate School of Engineering, Kyushu University, 744 Motooka, Nishi-ku, Fukuoka 819-0395, Japan; 8Institute of Quantum Life Science, National Institutes for Quantum and Radiological Science and Technology, Anagawa 4-9-1, Inage-ku, Chiba 263-8555, Japan

**Keywords:** nanowire structures, hydrothermal synthesis, mechanical-based rupture, antibacterial activity, cell compatibility

## Abstract

There are growing interests in mechanical rupture-based antibacterial surfaces with nanostructures that have little toxicity to cells around the surfaces; however, current surfaces are fabricated via top-down nanotechnologies, which presents difficulties to apply for bio-surfaces with hierarchal three-dimensional structures. Herein, we developed ZnO/SiO_2_ nanowire structures by using bottom-up approaches and demonstrated to show mechanical rupture-based antibacterial activity and compatibility with human cells. When *Escherichia coli* were cultured on the surface for 24 h, over 99% of the bacteria were inactivated, while more than 80% of HeLa cells that were cultured on the surface for 24 h were still alive. This is the first demonstration of mechanical rupture-based bacterial rupture via the hydrothermally synthesized nanowire structures with antibacterial activity and cell compatibility.

## 1. Introduction

During implant surgery, bacterial adhesion and growth on implanted material surfaces can lead to serious infectious diseases that threaten the life of transplant patients [1,2]. Bacterial adhesion is the most important step in the implant infection process and killing bacteria attached on surfaces is a promising way to prevent infection [2,3]. Composition-based antibacterial surfaces, such as silver [3,4], titanium dioxide [5] and polymer compounds [6,7,8,9,10], have been developed for conventional use in infection prevention. However, there are difficulties in applying currently available antibacterial surfaces due to their toxicity to cells around the surfaces, their inability to kill drug-resistant bacteria, and their lack of durability in a biological environment [11,12].

Recently, nanostructures on the wings of clanger cicadas (*Psaltoda claripennis*; *P. claripennis*) have been paid considerable attention due to their antibacterial activity that leads bacteria on the nanostructures to mechanically rupture [13,14]. Nanostructures on the wing of *P. claripennis* are roughly 60 nm in diameter, 200 nm in height and separated by 170 nm (Figure 1a) [13,14]. Bacteria attached on the nanostructure are mechanically ruptured in the following steps. Adhesion on nanostructures drastically increases the surface area of the bacteria cell membrane via the region suspended between nanostructures, the bacterial cell membrane is stretched, and the enough membrane stretching leads to bacterial rupture and inactivation [13,14,15]. It has been reported that this bacterial rupture is based on a mechanical mechanism using nanostructures and it is independent of surface components [14]. In addition to clanger cicada wings, dragonfly (*Diplacodes bipunctata*) wings also are able to rupture bacteria via the same mechanism [16]. These reports have promoted interest in the development of bio-inspired antibacterial surfaces [11,16,17,18].

As nanostructures inspired by clanger cicada wings, black silicon (bSi) nanowires were fabricated by reactive ion etching (RIE); these nanowires had a 20–80-nm diameter, 500 nm of height and 200–1800 nm of spacing [19]. When bacteria attached onto the bSi surface, they were ruptured mechanically. This report indicates that artificial nanostructures provide an antibacterial surface that is independent of the surface components and they have a potential for application to drug-resistant bacteria. Another study confirmed the bSi-based antibacterial surface ruptured bacteria without causing human cell inactivation [20]. A mechanical-based rupture is expected to be used by the biomaterial surface when embedded in a living body, such as implants. However, the bSi-based antibacterial surface is fabricated via top-down nanotechnologies, and the application of these nanotechnologies for biomaterials with hierarchal three-dimensional structures is difficult. Instead of the top-down technologies, bottom-up nanotechnologies allow us to synthesize organic [21] and inorganic [22,23,24,25] nanostructures in a self-assembled manner. Recent development of bottom-up nanotechnologies enabled the fabrication of nanostructures on surfaces with hierarchal 3-dimensional structures such as paper [26], fiber [24], meshes [27,28,29] and even bodies of organisms [26]. Therefore, for developing the biomaterial surfaces with bacterial resistance, the mechanical rupture-based antibacterial surface fabricated via the bottom-up nanotechnologies is desired.

In this work, we developed a bio-inspired antibacterial surface using nanowires that had a mechanical rupture-based antibacterial activity and compatibility with human cells (Figure 1b,c). The antibacterial surface had ZnO-SiO_2_ core-shell (ZnO/SiO_2_) nanowires that were fabricated by the hydrothermal synthesis of the ZnO core and followed by SiO_2_ deposition. The hydrothermal synthesis is a promising way to allow the nanowires to be fabricated on large and 3-dimensional biomaterial surfaces. The ZnO/SiO_2_ nanowires were 71 nm in diameter, 800–1200 nm in height, and approximately 110 nm of average spacing. The ZnO/SiO_2_ nanowires showed over 99% bacterial inactivation via incubating *Escherichia coli* for 24 h. Moreover, more than 80% of HeLa cells can be cultured on the ZnO/SiO_2_ nanowires for 24 h without rupture events. This is the first demonstration of mechanical rupture-based bacterial rupture on a cell-compatible ZnO/SiO_2_ nanowire surface fabricated by bottom-up approaches. We expect that this antibacterial surface will lead to the development of applications for 3-dimensional biomaterial surfaces.

## 2. Materials and Methods

### 2.1. Fabrication of ZnO Nanowire and ZnO/SiO_2_ Nanowire Substrates

Bare glass substrate (Matsunami Glass Industry, Ltd., Osaka, Japan), 50 × 50 mm in size for culturing bacteria, and fused silica substrate (Crystal Base Co., Ltd., Osaka, Japan), 12 mm in diameter for culturing human cells, were immersed into piranha solution, which was a mixture of hydrogen peroxide and sulfuric acid in a volume ratio of 1:4. The substrate was placed on a petri dish, the piranha solution was poured and the solution was heated at 180 °C on a hotplate for 2 h. After cooling, the substrate was taken away from the dish by using a Teflon tweezer, washed by Milli-Q water and heated at 300 °C to dry it by using an electric furnace (FB1314M, Thermolyne). The seed layer for ZnO nanowires was prepared by the following procedures (Appendix A) [30,31]. As seed solution, 75 mM zinc acetate dihydrate (Sigma-Aldrich Co., Ltd., Saint Louis, MO, USA) and 75 mM ethanolamine (Sigma-Aldrich Co., Ltd.) were dissolved into 2-methoxyethanol (Wako Pure Chemical Industries, Ltd., Osaka, Japan) at 65 °C. A total of 1000 or 100 µL of the seed solution was dropped on the bare glass substrate or fused silica substrate, respectively. Each substrate was spin-coated at 3000 rpm for 30 s, and then heated at 300 °C for 3 min. The above procedures were repeated six times to form the ZnO seed layer. Next, ZnO nanowires were hydrothermally synthesized (Appendix A). Each substrate with the ZnO seed layer was immersed into the nanowire growth solution, which was a mixture of 20 mM hexamethylenetetramine (Wako Pure Chemical Industry, Ltd.) and 20 mM zinc nitrate hexahydrate (Alfa Aesar) in Milli-Q water and heated at 95 °C for 5 h inside an oven (DKM400, YAMATO Scientific Co., Ltd., Tokyo, Japan). After washing with Milli-Q water, each substrate was dried by flowing nitrogen. A SiO_2_ layer (10 nm thick) was deposited on the ZnO nanowires using an atomic layer deposition apparatus (Savanna G2, Ultratech/Cambridge Nanotech Inc., Waltham, MA, USA). Tris(dimethylamino)silane (TDMAS; Japan Advanced Chemicals, Kanagawa, Japan) and ozone were used for SiO_2_ deposition at 150 °C with 100 cycles.

ZnO film, copper (Cu) film, which is generally known to have high cytotoxicity [32,33], and SiO_2_-deposited ZnO film were prepared on each substrate without nanowire structures. The ZnO film was fabricated by spin-coating where seed solution and a heating procedure were the same as in the ZnO nanowire fabrication described above. The SiO_2_-deposited ZnO film was prepared by SiO_2_ growth on the ZnO film by atomic layer deposition at 150 °C with 100 cycles. The Cu film (100 nm thick) was sputtered using a direct current sputtering apparatus (SC-701Mk Advance, Sanyu Electron Co., Ltd., Tokyo, Japan).

### 2.2. SEM Observation of ZnO Nanowires and ZnO/SiO_2_ Nanowires

We observed ZnO nanowires and ZnO/SiO_2_ nanowires using a field-emission scanning electron microscope (FESEM; SUPRA 40vp, Carl Zeiss, Oberkochen, Germany). Each observed sample was coated by 10-nm-thick gold deposited by a sputtering apparatus (MSP-mini, Vacuum Device, Ibaraki, Japan). The diameter of the nanowires and their density on a surface area were measured from SEM images using the image analysis software “Image J”.

### 2.3. Elemental Mapping of Nanowire Substrates and a Single ZnO/SiO_2_ Nanowire

The elemental mapping of ZnO nanowire substrate and ZnO/SiO_2_ nanowire substrate was performed using FESEM (JSM-7610F, Jeol, Tokyo, Japan) with the energy dispersive X-ray spectrometer (EDS). For cross-sectional images, a 15-nm-thick platinum layer was sputtered on the nanowire substrates and a 20-kV accelerating voltage was chosen. Elemental mapping of a single nanowire, which was peeled off from ZnO/SiO_2_ nanowire substrate, was performed without platinum sputtering and a 30-kV accelerating voltage was selected. Zn Kα (8.630 eV), Si Kα (1.739 eV) and O Kα (0.525 eV) were chosen for constructing each image.

### 2.4. Antibacterial Test

An antibacterial test for the evaluation of bacteria inactivation was performed based on the partially modified Japanese Industrial Standards (JIS Z 2801: 2010). The *Escherichia coli* (*E. coli*) DH5α were cultured in 2.5% LB medium (Sigma-Aldrich Co., Ltd.) at 37 °C for 48 h. The bacteria were washed by repeating the following procedures three times: the medium was centrifuged at 5000× *g* for 10 min, the supernatant was removed, and the bacteria were suspended into nutrient broth medium that had been diluted 500 times. The 500-fold diluted nutrient broth medium was prepared by dissolving 180 mg of nutrient broth medium (Nutrient broth, Eiken Chemical Co., Ltd., Tokyo, Japan) into 10 mL Milli-Q water, diluting 500 times and adjusting pH to 7.0–7.2 using sodium hydroxide solution. A bacteria-containing solution with 1.0 × 10^6^ cells/mL was prepared for the test.

Each nanowire substrate (ZnO and ZnO/SiO_2_) (50 × 50 mm in size) was immersed into ethanol for 10 min and dried by flowing nitrogen. Next, 250 µL of bacteria-containing solution was dropped on the substrate, and this was covered with a piece (40 × 40 mm in size) of ethanol-sterilized poly(ethylene) film (Asone Co., Osaka, Japan). Under a sufficiently high humidity environment, the bacteria were incubated at 35 °C for 24 h. The substrate and the poly(ethylene) film were washed out with 10 mL of soybean casein digest agar with lecithin, polysorbate 80 (SCDLP) culture medium (Eiken Chemical Co., Ltd.). The washout solution was collected and diluted 10 times by phosphate-buffered saline (PBS). The PBS was prepared by dissolving 3.4 g of potassium dihydrogenphosphate into 100 mL of sterilized Milli-Q water, adjusting the pH to 6.8–7.2 using sodium hydroxide solution, and diluting 800-fold with 0.85% NaCl solution (Wako Pure Chemical Industry, Ltd.). A total of 1 mL of washout solution diluted 10-fold and 15 mL of normal agar medium (Eiken Chemical Co., Ltd.) dissolved by heating at 48 °C were mixed in a sterile dish. After cooling by stirring the mixture, the bacteria were incubated at 35 °C for 48 h. We counted the number of colonies formed on the agar medium.

### 2.5. SEM Observation of Bacteria Cultured on Bare Glass Substrate and ZnO/SiO_2_ Nanowire Substrate

We observed *E. coli* cultured on bare glass substrate and ZnO/SiO_2_ nanowire substrate using FESEM. The *E. coli* were cultured on each substrate for 24 h using the 500-fold diluted nutrient broth medium. After washing the substrates with Milli-Q water, they were dried at room temperature for 24 h.

### 2.6. Fluorescence Microscopy Observation of HeLa Cells Cultured on Various Substrates

HeLa cells were employed for addressing cell viability, which was used as a cellular model for evaluating the cell compatibility of material surfaces toward implant applications [34]. HeLa cells (1.5 × 10^5^ cells/mL) were cultured at 37 °C for 24 h on bare glass substrate, ZnO film substrate, ZnO nanowire substrate, SiO_2_-deposited ZnO film substrate, and ZnO/SiO_2_ nanowire substrate. Culturing the cells on the SiO_2_-deposited substrate was not demonstrated due to showing no antibacterial activity. The HeLa cells were cultured in minimum essential medium (MEM, Sigma-Aldrich Co., Ltd.) with 10% (*v*/*v*) inactivated fetal bovine serum (Bio-West Inc., Nuaillé, France), 1% (*v*/*v*) penicillin/streptomycin (Life Technologies Co.) and 1% (*v*/*v*) amino acid (100 × non-essential amino acids for MEM eagle, MP Biomedicals, LLC.). A total of 150 µL of PBS (Life Technologies Co., Carlsbad, CA, USA) with 2 µM calcein-acetoxymethyl (AM; Thermo Fisher Scientific Inc., Waltham, MA, USA) and 4 µM propidium iodide (PI; Dojindo Molecular Technologies, Inc., Kumamoto, Japan) were added to the cultured cells and they were incubated at 37 °C under 5% CO_2_ condition for 15 min. Using a confocal microscope (TCS-STED CW, Leica Microsystems, Wetzlar, Germany), we observed live and dead cells stained by calcein-AM and PI, respectively.

### 2.7. Cell Viability Measurement Using a Flow Cytometer

HeLa cells (2.5 × 10^5^ cells/mL) were cultured for 24 h on bare glass substrate, ZnO film substrate, ZnO nanowire substrate, SiO_2_-deposited ZnO film substrate, ZnO/SiO_2_ nanowire substrate, and Cu film substrate. The culture medium was recovered for the following cell viability measurement. The cells on each substrate were washed with 300 µL of PBS (Sigma-Aldrich Co., Ltd.) and the PBS solution was collected. 10 × trypsin solution (Sigma-Aldrich Co., Ltd.) was added to the well and the cells were incubated for 10 min to suspend those cells attached on each substrate. The trypsin solution was collected and mixed with the recovered culture medium and the PBS. The mixed solution was centrifugated at 180× *g* for 3 min and the supernatant was removed. The cells were resuspended into PBS to adjust them to a concentration of 3.0 × 10^5^ cells/mL. A total of 50 µL of the cell suspension and 450 µL of fluorescent reagent (Muse Count and Viability Assay, Merck Millipore Co., Burlington, MA, USA) were mixed together. This fluorescent reagent contained two types of dye: one for staining dead cells only and the other for staining both live and dead cells. After incubating for 5 min, cell viability was measured using a flow cytometer (Cell analyzer, Merck Millipore Co.).

### 2.8. SEM Observation of HeLa Cells Cultured on ZnO/SiO_2_ Nanowire Substrate

HeLa cells were cultured on ZnO/SiO_2_ nanowire substrate for 24 h. After washing the substrate with Milli-Q water, the substrate was dried at room temperature for 24 h. We observed the HeLa cells with FESEM.

## 3. Results and Discussion

### 3.1. Fabrication of ZnO/SiO_2_ Nanowires for the Antibacterial Surface

*P. claripennis* wing-inspired ZnO/SiO_2_ nanowires were fabricated via hydrothermal synthesis (Figure 2a,b). ZnO nanowires and ZnO/SiO_2_ nanowires were 51.0 ± 8.2 nm and 71.7 ± 12.1 nm in diameter, respectively, and approximately 110 nm of average spacing of ZnO/SiO_2_ nanowires was calculated from 83 ± 5 nanowires/µm^2^ of the surface area density. From the average diameter, we estimated that the SiO_2_ layer was deposited with a thickness of about 10 nm. Nanostructures on *P. claripennis* wing are 60 nm in diameter and spaced 170 nm apart from center to center [13,14], which were similar to the fabricated nanowire sturectures. From elemental mapping images, ZnO nanowires did not have signals derived from Si Kα (Figure 2(ci)); on the other hand, ZnO/SiO_2_ nanowires had signals from Si (Figure 2(cii)). We confirmed nanowire structures had Si-derived signals from the bottom to the tip of the nanowire structures, and therefore, ZnO/SiO_2_ nanowires were fully covered with the SiO_2_ layer. Single nanowire observation also revealed that our nanowires had a structure with the ZnO core and SiO_2_ shell (Figure 2(ciii) and Appendix A).

### 3.2. Antibacterial Tests

Antibacterial activities of SiO_2_ film and ZnO/SiO_2_ nanowire substrates were evaluated by the partially modified JIS Z 2801: 2010. *E. coli* were cultured on substrates with three types of silicon oxides (SiO_x_)-based surface, i.e., bare glass substrate, SiO_2_ film substrate and ZnO/SiO_2_ nanowire substrate; and two types of ZnO-based surface, i.e., ZnO film substrate and ZnO nanowire substrate. The bacteria cultured on each substrate before and after washing out were observed (Figure 3a). From these fluorescence images, we confirmed that the bacteria on each substrate could be washed out and collected for the colony counting. However, in the case of the ZnO film substrate and the ZnO nanowire substrate, bacteria attached on their surfaces via electrostatic interaction (Appendix A), which was originating from the positive ZnO surface (present around pH 7.0) and the negative bacteria surface. Therefore, we could not perform quantitative analysis due to the difficulty in application of the partially modified JIS-based antibacterial test to ZnO film and ZnO nanowire substrates. After the washing out of bacteria from the SiO_x_-based surfaces, the recovered *E. coli* were cultured on an agar medium mixed with washout solution for 24 h in a sterile dish, and then, the number of colonies formed on the agar medium was counted. We confirmed no colonies were formed on the agar medium when the ZnO/SiO_2_ nanowire substrate was used. Antibacterial activity (R) against bare glass substrate was calculated from the number of the counted colonies by using the following Equation (1):(1)$$R=\left\{\log\left(\frac{B}{A}\right)-\log\left(\frac{C}{A}\right)\right\}=\log\left(\frac{B}{C}\right)$$

A, B and C are the average counts of colonies formed on the agar medium just after incubation, after culturing on bare glass substrate for 24 h, and after culturing on SiO_2_ film substrate or ZnO/SiO_2_ nanowire substrate for 24 h, respectively. We can see that the ZnO/SiO_2_ nanowire substrate had a very high antibacterial activity value, while SiO_2_ film substrate did not show antibacterial activity (Figure 3b). Since these results clearly indicate that the SiO_2_ component would have little antibacterial activity against *E. coli*, the antibacterial activity of ZnO/SiO_2_ nanowire substrate was based on a mechanical rupture mechanism using the nanowire structures. According to JIS Z 2801: 2010, an antibacterial activity value over 2 observed on the ZnO/SiO_2_ nanowire substarate means the nanowire structures could inactivate over 99% of the bacteria. Thus, the ZnO/SiO_2_ nanowire substrate was an effective antibacterial surface.

From the SEM observation, we confirmed rod-shaped *E. coli* were on the bare glass substrate (Figure 3(ci)). Whereas, only debris of bacterial cells were observed on the ZnO/SiO_2_ nanowire substrate (Figure 3(cii)) and some debris of ruptured bacteria were washed out during the washing procedure. From SEM images, we concluded *E. coli* were ruptured via a mechanical mechanism.

A bacteria membrane has an adhesion energy which leads to membrane attachment on nanowire structures before finally undergoing mechanical rupture via membrane stretching. In this work, we attained the nanowire-induced mechanical rupture of bacteria; this was consistent with several reports [11,19,20,35]. Simulations of mechanically rupturing bacteria showed that a high density of nanowires (> 40 nanowires/µm^2^), with not too small a diameter (> 30 nm) and an appropriate length, (> 200 nm) was suitable for bacteria rupture [15]. Another study demonstrated the efficiency of bacterial lysis with respect to the density of nanostructures on cicada wings and showed that the wing with a high density of nanostructures had more efficient bacteria lysis in comparison to that with a low density of nanostructures [36]. Our present surface had 83 nanowires/µm^2^ and they were 71 nm in diameter and 800–1200 nm in length, which satisfied the aforementioned nanostructure conditions. Therefore, it was thought that the present nanowires showed a highly effective antibacterial property.

### 3.3. Cell Viability

Cell viability was measured for HeLa cells cultured on bare glass substrate, ZnO film substrate, ZnO nanowire substrate, ZnO/SiO_2_ nanowire substrate, SiO_2_-deposited ZnO film substrate, and Cu film substrate. The HeLa cells cultured on each substrate were observed with the fluorescent microscope (Figure 4a). The cells cultured on each substrate were collected and quantitative cell viability analysis was performed using the flow cytometer (Figure 4b). Cell viability when cultured on bare glass substrate was defined as 100% and cell viability when cultured on another substrate was calculated, accordingly. We confirmed that the ZnO/SiO_2_ nanowire substrate had more than 80% cell viability, which was equivalent to that of the SiO_2_-deposited ZnO film substrate. ZnO film, ZnO nanowire and Cu film substrates had cell viability of 90, 61 and 11%, respectively. Since Cu film is generally known to have high cytotoxicity [32,33], this value was reasonable.

We observed that the ZnO nanowire substrate had many dead cells stained by PI (Figure 4a), which means that the ZnO nanowires had high cytotoxicity caused by elution of zinc ions [37,38]. Since the ZnO nanowire substrate had over a 20 times larger surface area than the ZnO film substrate, this led to a large amount of eluted zinc ions in the biological fluids [37,38,39]. Therefore, ZnO nanowire substrate showed high cytotoxicity. Due to suppressing the elution of zinc ions by the SiO_2_ shell layer, the ZnO/SiO_2_ nanowire showed high cell viability. Furthermore, the viability of cells cultured on the ZnO/SiO_2_ nanowire substrate was almost the same as that on the SiO_2_-deposited ZnO film substrate. Thus, nanostructures have only a small effect on cell viability [40].

The SEM observation revealed that the membranes of HeLa cells cultured on ZnO/SiO_2_ nanowire substrate had a few penetrations by the nanowires and the extent of membrane stretching was small (Figure 4c). This result enabled to judge that human cells could be cultured on the ZnO/SiO_2_ nanowire. The small deformation of the cell membrane on the nanowires may be due to the focal adhesions which are key structural components to anchor an actin cytoskeleton on the surfaces [40,41]. Human cells could adhere on the tip of nanostructures via formation of focal adhesions [42,43]. Therefore, a high density of nanowire structures can provide a lot of focal adhesion sites to the cell membrane at the nanowire tips [44,45]. The deformation of the cell membrane must be effectively suppressed because higher energy is necessary to deform the cell membrane under the larger number of anchoring points [46,47,48]. This can be the reason why the cells could survive without sinking down towards the underside of the nanowires. This consideration is supported by the behavior of erythrocytes on nanostructures. Erythrocytes that are non-adhesive human cells are inactivated via the nanowire-induced membrane stretching followed by mechanical-based rupture [49]. It is considered that non-adhesive cell membrane cannot make a tight anchoring structure on the nanowire tip, so the cell deformation, leading to the rupture, can proceed. These mechanisms for adhesions of human cells on nanostructures are not fully known [17]; however, our results are consistent with knowledge obtained so far.

## 4. Conclusions

In this work, we developed a *P. claripennis* wing-inspired nanowire structure formed by the hydrothermal synthesis of the ZnO core and follow-by SiO_2_ shell deposition. This nanowire surface used a mechanical rupture mechanism to obtain an antibacterial rate of 99% against *E. coli*. Moreover, 80% of human cells were alive when cultured on the nanowire surface, which was originated from adhesive properties of cells. Nanowire structures fabricated by bottom-up approaches (hydrothermal synthesis and followed by ALD deposition) was demonstrated to show mechanical rupture-based antibacterial activity and cell compatibility. We expect that it can be applied to get 3-dimensional biomaterials for implant surgery.

## Figures and Tables

**Figure 1 micromachines-11-00610-f001:**
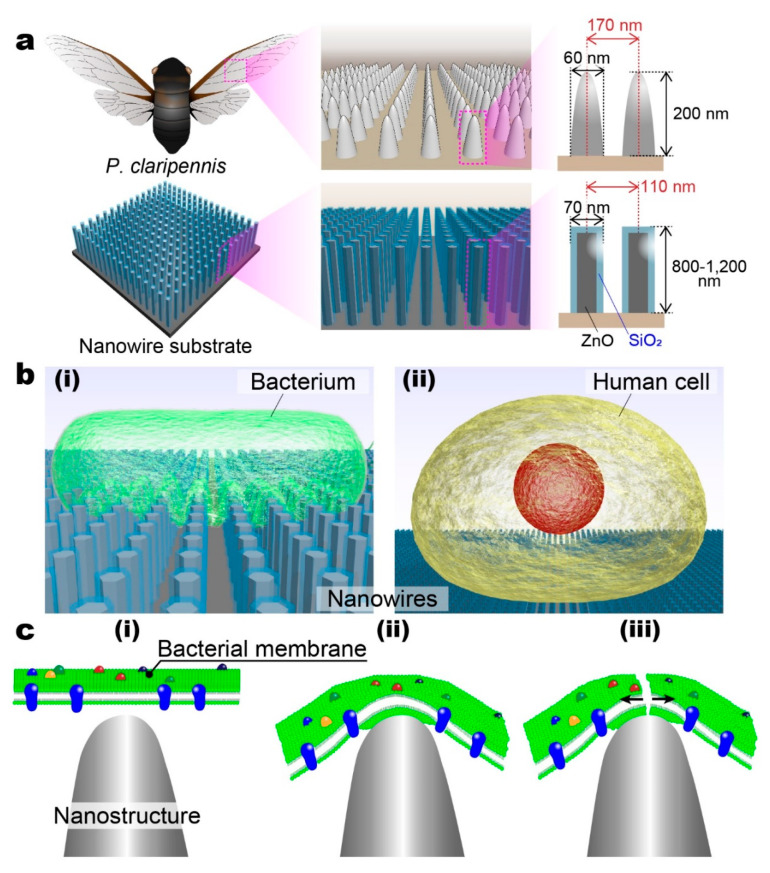
(**a**) Schematic illustrations of nanostructures on the wing of *P. claripennis* (upper) and antibacterial nanowire substrate (lower). Although arraigned nanowire structures are simply illustrated here, fabricated nanowires were randomly orientated. (**b**) Schematic illustrations of (i) a bacterium and (ii) human cell on nanowire structures. While the human cell attached on the nanowire structures was alive, the bacterium on the structures was mechanically ruptured and inactivated. (**c**) Schematic illustrations of the bacterium rupture mechanism. (i) A cell membrane of the bacterium was attached on the nanostructure. (ii) The membrane was stretched via the nanostructure and the surface area of the membrane increased. (iii) The membrane was ruptured due to too much stretching.

**Figure 2 micromachines-11-00610-f002:**
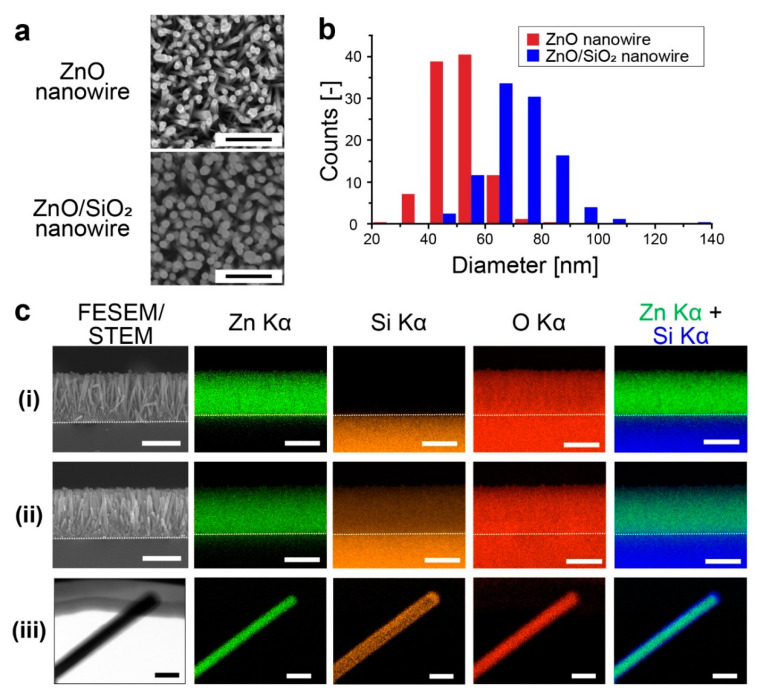
(**a**) Scanning electron microscope (SEM) images of ZnO nanowire (upper) and ZnO/SiO_2_ nanowire (lower) as a top view (scale bars, 500 nm). (**b**) Nanowire diameter of ZnO nanowires (red) and ZnO/SiO_2_ nanowires (blue). (**c**) Elemental mapping images by field-emission scanning electron microscope (FESEM) or scanning transmission electron microscope (STEM); Zn Kα, Si Kα, O Kα, and merged. Merged images were constructed from Zn Kα (green) and Si Kα (blue). Mapping images are for: (i) free-standing ZnO nanowires (scale bars, 1 µm), (ii) free-standing ZnO/SiO_2_ nanowires (scale bars, 1 µm) and (iii) single ZnO/SiO_2_ nanowire (scale bars, 100 nm).

**Figure 3 micromachines-11-00610-f003:**
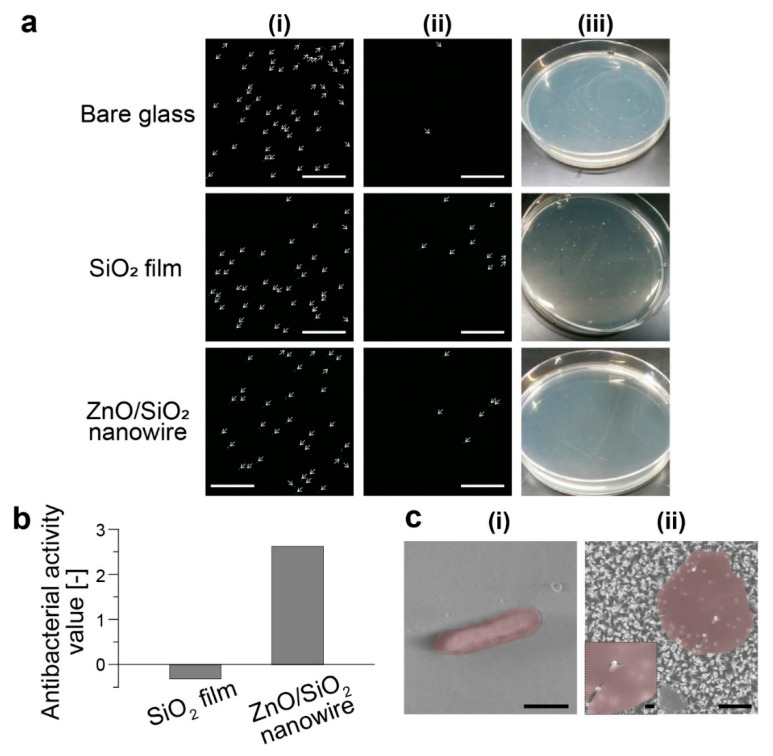
(**a**) Counting viable bacteria cultured on each substrate. (i,ii) Fluorescence images of bacteria on each substrate (i) before and (ii) after washing out (scale bars, 100 µm). (iii) Photographs of an agar medium with colonies. Bacteria collected from each substrate were cultured on an agar medium to count colonies. (**b**) Antibacterial activity value for each substrate. Antibacterial activity value was normalized by that of the bare glass substrate. (**c**) SEM images of a bacterium cultured on (i) bare glass substrate (scale bar, 1 µm) and (ii) ZnO/SiO_2_ nanowire substrate (scale bars, 1 µm and 200 nm (insertion)).

**Figure 4 micromachines-11-00610-f004:**
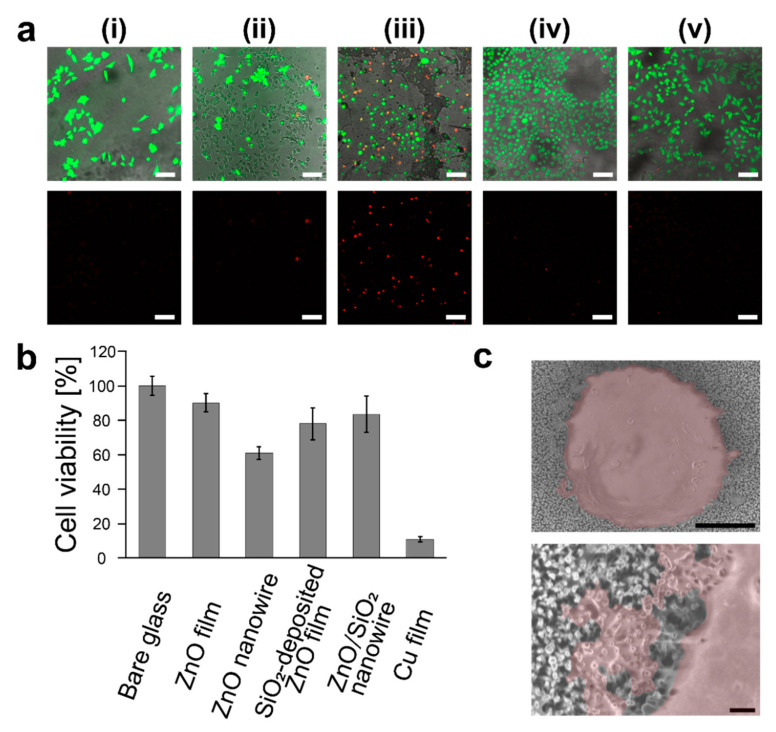
(**a**) Fluorescence images of HeLa cells cultured on various substrates: (i) bare glass, (ii) ZnO film, (iii) ZnO nanowires, (iv) SiO_2_-deposited ZnO film, (v) ZnO/SiO_2_ nanowires (upper, merged images of cells stained by calcein-acetoxymethyl (AM) and PI (propidium iodide); lower, fluorescence images of cells stained by PI; scale bars, 100 µm). (**b**) Cell viability of HeLa cells cultured on each substrate. These viabilities were measured by using flow cytometry and viability of the cells on bare glass substrate was defined as 100%. (**c**) SEM images of HeLa cells cultured on ZnO/SiO_2_ nanowire substrate (scale bars, 5 µm (upper) and 500 nm (lower)).

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
