# Peer review of "Mechanical Rupture-Based Antibacterial and Cell-Compatible ZnO/SiO2 Nanowire Structures Formed by Bottom-Up Approaches"

_micromachines, 2020, doi:10.3390/mi11060610_

Round 1

Reviewer 1 Report

The manuscript  reproted  a method about synthesising of the ZnO core and follow‐by SiO2 shell deposition.The topic  is very interesting,but there are some drawbacks in the manuscript.So  I  suggest the manuscript should be accepted afer major revisions.

1.The author should revise the introduction section,the related works is missing.Compared the related work"MATERIALS LETTERS,2015,160:218-221;MATERIALS LETTERS,2012,67(1):8-10;CHEMICAL ENGINEERING JOURNAL,2020,390:124522,the author should emphasis the novety of the manuscript.

2.The antibacterial mechanism is too simple.The authors  should add more details to the manuscript.

3.What is the fastness of the Cell-compatible Nanowire Structures?After washing several times,Do you confirm it ?

4. In the conclusion section,the authors calimed "This is the first demonstration of a mechanical‐based antibacterial and cell compatible nanowire surface fabricated by hydrothermal synthesis".The author should 

confirm it.There are a lot of related work,So it may be a mistake.

Author Response

General comments

The manuscript  reproted a method about synthesising of the ZnO core and follow‐by SiO2 shell deposition. The topic is very interesting, but there are some drawbacks in the manuscript. So I suggest the manuscript should be accepted after major revisions.

Reply to general comments

We would like to greatly thank Reviewer 1 for the fairly favorable evaluation of our manuscript.  As the reviewer commented, some revisions should be needed for publication in Micromachines.  We acknowledge these important suggestions, and carefully considered them in making the revisions as follows.

Comment 1

The author should revise the introduction section, the related works is missing. Compared the related work"MATERIALS LETTERS,2015,160:218-221;MATERIALS LETTERS, 2012, 67(1): 8-10; CHEMICAL ENGINEERING JOURNAL, 2020, 390:124522, the author should emphasis the novety of the manuscript.

Reply to major comment 1

We would like to greatly thank for your suggestion about bottom-up technologies.  We added the following statement and inserted references: “Instead of the top-down technologies, bottom-up nanotechnologies allow to synthesize organic (Macromolecules, 2019, 52, 6318-6329) and inorganic (Sci. Rep., 2017, 7, 4155; Chemical Engineering Journal, 2020, 390; Materials Letters, 2012, 67, 8-10; Materials Letters, 2015, 160, 218-221) nanostructures in a self-assembled manner.  Recent development of bottom-up nanotechnologies enabled fabrication of nanostructures on surfaces with hierarchal 3-dimensional structures such as paper, fiber (Materials Letters, 2012, 67, 8-10), meshes and even bodies of organisms.”

Comment 2

The antibacterial mechanism is too simple. The authors should add more details to the manuscript.

Reply to major comment 2

We would like to greatly thank for your suggestion.  As you mentioned, the mechanism of mechanical rupture seems to be simple, however, the mechanism currently has a lack of fully being elucidated.  Although some theoretical models are suggested, mathematical explanation in the models are beyond this paper.  Thus, we modified the mechanism as the following with inserting references: “Adhesion on nanostructures drastically increases the surface area of the bacteria cell membrane via the region suspended between nanostructures, the bacterial cell membrane is stretched, and the enough membrane stretching leads to bacterial rupture and inactivation (Biophys. J., 2013, 104, 835; Small 2012, 8, 2489; Phys. Chem. Chem. Phys., 2016, 18, 1311)”

Comment 3

What is the fastness of the Cell-compatible Nanowire Structures?After washing several times,Do you confirm it ?

Reply to major comment 3

We would like to thank for your question.  The nanowire structures could be seen clearly in Figure 3 and 4 which are SEM images after washing, thus, there is no structural problem.

Comment 4

In the conclusion section, the authors calimed "This is the first demonstration of a mechanical-based antibacterial and cell compatible nanowire surface fabricated by hydrothermal synthesis". The author should confirm it. There are a lot of related work. So it may be a mistake.

Reply to major comment 4

We would like to very thank for your constructive comments.  As you mentioned, there is a report on a surface with antibacterial and cell compatible nanostructures which was fabricated under the harsh conditions (240 °C, 1 M NaOH; Sci. Rep., 2015, 4, 7122).  Additionality, the process was hydrothermally etching.  On the other hand, our method can mildly synthesize nanowires (95 °C, pH ~7), thus, applying to biomaterial can be realized.  Although some optimizations on cleaning (Piranha solution) and drying (heating at 300 °C) substrates are required, they can be replaced with other methods.  We modified the statement as “Nanowire structures fabricated by bottom-up approaches (hydrothermal synthesis and followed-by ALD deposition) was demonstrated to show mechanical rupture-based antibacterial activity and cell compatibility.”

Reviewer 2 Report

Shimada et al. report on the development of an anti-bacterial surface based on ZnO/SiO2 nanowires. The authors show the inactivation of bacteria seeded on this surface induced by mechanical rupture, while the HeLa cell viability did not seem to be affected. Overall, the manuscript is written very well with clear explanations and proper citation of the relevant work. Figures are also clear and of high quality. However, there are a few points that need to be addressed:

  1. To demonstrate the anti-bacterial efficacy of the nanowires, a Gram-negative strain was used (E. coli). Testing the anti-bacterial function also on Gram-positive bacteria will be necessary for a better evaluation of nanowires as anti-bacterial surfaces. Gram-positive bacteria have a much thicker peptidoglycan layer, which is likely to influence the efficiency of mechanical rupture.
  2. As a comparison, the authors studied the viability of HeLa cells on the nanowire surface. However, this cell line does not represent a 'normal' healthy cell population. It is known that cancer cells significantly differ from their healthy counterparts both mechanically and physiologically. Several studies showed that cancer cells are somehow softer and they can deform without mechanical rupture. Therefore, it is necessary to show the viability of a different adherent cell type as well (e.g. fibroblasts).
  3. Regarding the fabrication of nanowires, a discussion can be added about how to modify the diameter and spacing of nanowires, and how different nanowire dimensions would influence the resulting anti-bacterial performance.
  4. To better characterize the nanowires, as well as the other control substrates shown in Figure 2, surface roughness and contact angle measurements would be interesting. Surface roughness is known to have a direct impact on cell adhesion and proliferation. Contact angle measurements would give additional information on surface hydrophilicity, which also influences cell-substrate interactions.
  5. In the conclusion part, the authors propose that these structures can be applied in implants, however, they also state that these nanostructures cause erythrocyte rupture (Line 298). I believe this is the major drawback of the system, which shows that these nanowires probably may not be used inside the body. The authors link the reason for erythrocyte inactivation to non-adherent nature of these cells. Then, would all other non-adherent cells (e.g. lymphocytes) would be inactivated (ruptured?) as well? I think this point (as well as other potential drawbacks) should be mentioned and discussed further. 

Author Response

General comments

Shimada et al. report on the development of an anti-bacterial surface based on ZnO/SiO2 nanowires. The authors show the inactivation of bacteria seeded on this surface induced by mechanical rupture, while the HeLa cell viability did not seem to be affected. Overall, the manuscript is written very well with clear explanations and proper citation of the relevant work. Figures are also clear and of high quality. However, there are a few points that need to be addressed:

Reply to general comments

We would like to appreciate Reviewer 2 for the constructive comments.  As the reviewer commented, some points should be modified for publication in Micromachines.  We acknowledge these important suggestions, and carefully considered them in making the revisions as follows.

Comment 1

To demonstrate the anti-bacterial efficacy of the nanowires, a Gram-negative strain was used (E. coli). Testing the anti-bacterial function also on Gram-positive bacteria will be necessary for a better evaluation of nanowires as anti-bacterial surfaces. Gram-positive bacteria have a much thicker peptidoglycan layer, which is likely to influence the efficiency of mechanical rupture.

Reply to major comment 1

We greatly thank you for these important suggestions.  As you mentioned, gram-positive bacteria have much thicker peptidoglycan layer, thus, they are expected to be more rigid.  However, in the reference (Nat. Commun., 2013, 4, 2838), artificial nanowire structures could mechanically inactivate Gram-negative (Pseudomonas aeruginosa) and Gram-positive (Staphylococcus aureus) bacteria with the almost same level.  Thus, we expect that our nanowires also have a potential to mechanically rupture both of Gram-negative and Gram-positive bacteria.  In our future work for applying the nanowires developed here to 3-dimentional substrate, the efficiency of mechanical rupture against Gram-negative and Gram-positive bacteria must be demonstrated.

Comment 2

As a comparison, the authors studied the viability of HeLa cells on the nanowire surface. However, this cell line does not represent a 'normal' healthy cell population. It is known that cancer cells significantly differ from their healthy counterparts both mechanically and physiologically. Several studies showed that cancer cells are somehow softer, and they can deform without mechanical rupture. Therefore, it is necessary to show the viability of a different adherent cell type as well (e.g. fibroblasts).

Reply to major comment 2

We greatly appreciate for your important suggestions.  This work is motivated to demonstrate that nanowires fabricated via bottom-up technologies enable both to inactivate bacteria and have cell compatibility.  HeLa cell is employed to prove our concept.  However, as you mentioned, HeLa cells are one of abnormal cell and show different mechanical and physiological properties with comparison to normal healthy cell population.  Previously, other group showed that fibroblast cells cultured on artificial nanowires, which were fabricated top-down technologies, could be cultured, thus, nanowires show compatibility with “normal” healthy cells (ACS Appl. Mater. Interfaces, 2016, 8, 22025-22031).  The cell compatibility was also confirmed on nanowires fabricated via bottom-up technologies (Npg Asia Mater., 2016, 8, e249).  Thus, our nanowires can be applied to “normal” healthy cells.  In the future work, we plan to demonstrate to culture cells on 3-dimentional substrate with our nanowires toward application of implant devices, “normal” healthy cells must be evaluated.

Comment 3

Regarding the fabrication of nanowires, a discussion can be added about how to modify the diameter and spacing of nanowires, and how different nanowire dimensions would influence the resulting anti-bacterial performance.

Reply to major comment 3

The aim of this study is the development of surfaces showing antibacterial and cell-compatibility.  Thus, we employed strategies for obtaining an efficient antibacterial surface as mentioned in the discussion part as “high density of nanowires (> 40 nanowires/µm2), not too small diameter (> 30 nm) and appropriate length (> 200 nm) were suitable for bacteria rupture”.  Based on the strategies, the diameter and spacing of nanowires were controlled via modifying growth conditions as the thickness of seed layer and the concentration of nanowire growth solution.

Comment 4

To better characterize the nanowires, as well as the other control substrates shown in Figure 2, surface roughness and contact angle measurements would be interesting. Surface roughness is known to have a direct impact on cell adhesion and proliferation. Contact angle measurements would give additional information on surface hydrophilicity, which also influences cell-substrate interactions.

Reply to major comment 4

We greatly appreciate for your important suggestions.  As you mentioned, water contact angle is effective information on cell-substrate interactions.  Previously, we measured water contact angle of slide glass with ZnO and SiO2 films, which showed 25.6 ± 4.3 and 4.4 ± 1.0 degree, respectively (data is not provided).  Thus, ZnO/SiO2 nanowires showed relatively high cell viability because its outer layer was hydrophilic SiO2.

Comment 5

In the conclusion part, the authors propose that these structures can be applied in implants, however, they also state that these nanostructures cause erythrocyte rupture (Line 298). I believe this is the major drawback of the system, which shows that these nanowires probably may not be used inside the body. The authors link the reason for erythrocyte inactivation to non-adherent nature of these cells. Then, would all other non-adherent cells (e.g. lymphocytes) would be inactivated (ruptured?) as well? I think this point (as well as other potential drawbacks) should be mentioned and discussed further.

Reply to major comment 5

We would like to thank the reviewer for the question.  Since lymphocytes are circulating in blood, we expect rupture events are not be happened frequently due to the fact that erythrocytes should be incubated at least three minutes for mechanical rupture-based inactivation.

Reviewer 3 Report

First, the paper by Shimada et al. reports interesting results, is concise, is well written and the work has been well conducted. I would like to congratulate the authors for this nice work. Some issues (unclarities and addition of some further comments) should be addressed to improve the manuscript, and make it even more accessible to the wide readership it has the potential to achieve. Especially with respect to the experimental section, some further clarifications are required. Therefore, I recommend major revisions. In the following, I’m listing my concerns/questions/comments.

  • I suggest that both the title and the abstract should contain the type of material (chemical type), i.e. ZnO/SiO2 nanowire structures. For me it took quite a while until I found in the text which chemical nature the reported nanowires have. I believe this should be apparent for the readers earlier.
  • “Mechanical-based” vs. “mechanical-rupture based” are used interchangeably in the manuscript, and I think they should not. “Mechanical based” (e.g. used twice in the abstract) can me anything mechanical (flexibility of compressibility are also mechanical features). I recommend to opt for choosing the clearer term (in my opinion: mechanical rupture-based, between these two) in using it throughout the manuscript
  • Again regarding “mechanical rupture based”: Although the term is better than “mechanical-based”, it took until in the main text, that it is the bacteria that rupture (and not the nanowires). With title and abstract alone, I had thought that the ZnO wires break, and these spikes then de-activate the bacteria. I then understood that this is not the case. The fact that the bacteria rupture without the nanostructured surfaces disintegrating is great, since it makes the surfaces potentially re-usable (or better: potentially useable over a long time). This is a great asset of the reported materials, and therefore I would recommend that the authors stress this even further.
  • Could the authors briefly comment in the manuscript why they used HeLa cells over any other common cell line? “HeLa cells were chosen for….”
  • The authors nicely introduce the biological appearance and functioning of antibacterial nanostructures on insect wings (p.2). There is two issues that I think need further commenting on, especially as pertinent to the authors’ biomimetic materials: (i) In Fig.1, the natural nanostructures are basically nanowires arranged in a quadratic pattern. While the authors make it clear, that the mode of action of the nanostructures in living nature is independent of the type of material used (p.2, l.51), I think the authors should comment on the pattern that is formed (especially, since their nanowires are not arranged in a quadratic pattern; cf. Fig.2, p.6). (ii) It would be helpful to comment on the size of bacteria vs. human cells, as well as the thickness and composition of bacterial membranes vs. human cell membranes. From Fig.1, it appears that to human cells such a nanowire structuring “feels” like a mere surface roughness, while it ‘feels’ like huge obstacles to bacteria.
  • 3: The nice introduction to biological systems should be completed by a short summary of hydrothermal nanostructuring, especially since bottom-up techniques are generally mentioned (p.3, l.72 and following lines). While it is indeed rare (compared to the hydrothermal synthesis of nanoobjects, such as nanoparticles), hydrothermal nanostructuring of surfaces has been reported for both inorganic materials (https://doi.org/10.1038/s41598-017-04395-0), and organic materials (https://pubs.acs.org/doi/abs/10.1021/acs.macromol.9b00985). I recommend that the authors comment on that.
  • Experimental part, p3: (i) spaces between number and unit (300_˚C, etc); (ii) I’d urge the authors to provide more information on the piranha treatment, because the substrates were heated in piranha solution to 180 degC (it could be dangerous for other researchers to reproduce this approach without further instructions, e.g.: which apparatus was used? Flask plus reflux condenser? Autocave?); (iii) p.3, l.100: 300 degC for 3min (which oven was used? How could 300 degC be reached so quickly?); (iv) l. 102 and following lines: In which apparatus was the HT treatment performed?
  • Characterization: XRD of the nanostructures is necessary. As per SEM, the Zn structure looks like hexagonal wurtzite, but this needs to be proven through XRD. Also, I suppose that the SiO2 layer is amorphous (but this has to be shown through XRD).
  • Motivation: Why were the ZnO nanowires coated with SiO2? 

Author Response

General comments

First, the paper by Shimada et al. reports interesting results, is concise, is well written and the work has been well conducted. I would like to congratulate the authors for this nice work. Some issues (unclarities and addition of some further comments) should be addressed to improve the manuscript, and make it even more accessible to the wide readership it has the potential to achieve. Especially with respect to the experimental section, some further clarifications are required. Therefore, I recommend major revisions. In the following, I’m listing my concerns/questions/comments.

Reply to general comments

We would like to thank Reviewer 3 for the fairly favorable evaluation of our manuscript and make it acceptable for publication in Micromachines.  We would like to reply to the comments and suggestions specifically as follows.

Comment 1

I suggest that both the title and the abstract should contain the type of material (chemical type), i.e. ZnO/SiO2 nanowire structures. For me it took quite a while until I found in the text which chemical nature the reported nanowires have. I believe this should be apparent for the readers earlier.

Reply to major comment 1

We would like to greatly thank your suggestion.  We changed the title as “Mechanical Rupture-based Antibacterial and Cell-compatible ZnO/SiO2 Nanowire Structures Formed by Bottom-up Approaches”, and added material information in abstract as “Herein, we developed ZnO/SiO2 nanowire structures by using bottom-up approaches and demonstrated to show mechanical-rupture based antibacterial activity and compatibility with human cells.”  Additionally, the conclusion part was modified as “Nanowire structures fabricated by bottom-up approaches (hydrothermal synthesis and followed-by ALD deposition) was demonstrated to show mechanical rupture-based antibacterial activity and cell compatibility.”

Comment 2

“Mechanical-based” vs. “mechanical-rupture based” are used interchangeably in the manuscript, and I think they should not. “Mechanical based” (e.g. used twice in the abstract) can me anything mechanical (flexibility of compressibility are also mechanical features). I recommend to opt for choosing the clearer term (in my opinion: mechanical rupture-based, between these two) in using it throughout the manuscript

Reply to major comment 2

We thank you for your suggestion.  As you mentioned, we revised “mechanical-based” into “mechanical-rupture based”.

Comment 3

Again regarding “mechanical rupture based”: Although the term is better than “mechanical-based”, it took until in the main text, that it is the bacteria that rupture (and not the nanowires). With title and abstract alone, I had thought that the ZnO wires break, and these spikes then de-activate the bacteria. I then understood that this is not the case. The fact that the bacteria rupture without the nanostructured surfaces disintegrating is great, since it makes the surfaces potentially re-usable (or better: potentially useable over a long time). This is a great asset of the reported materials, and therefore I would recommend that the authors stress this even further.

Reply to major comment 3

We would like to greatly thank for your suggestions.  Previously, we confirmed the nanowire structures still existed after heating treatment at 600 °C which enables to remove organic contaminants derived from bacteria and cells from the substrate, thus they can potentially reusable.

Comment 4

Could the authors briefly comment in the manuscript why they used HeLa cells over any other common cell line? “HeLa cells were chosen for….”

Reply to major comment 4

We thank you for your suggestion.  We added our comment on the reason why HeLa cells were used as the following: “HeLa cells were employed for addressing cell viability, which was used as cellular model for evaluating cell compatibility of material surfaces toward implant applications (J. Mater. Sci.: Mater. Med., 2014, 25, 1425-1434).”

Comment 5

The authors nicely introduce the biological appearance and functioning of antibacterial nanostructures on insect wings (p.2). There is two issues that I think need further commenting on, especially as pertinent to the authors’ biomimetic materials: (i) In Fig.1, the natural nanostructures are basically nanowires arranged in a quadratic pattern. While the authors make it clear, that the mode of action of the nanostructures in living nature is independent of the type of material used (p.2, l.51), I think the authors should comment on the pattern that is formed (especially, since their nanowires are not arranged in a quadratic pattern; cf. Fig.2, p.6). (ii) It would be helpful to comment on the size of bacteria vs. human cells, as well as the thickness and composition of bacterial membranes vs. human cell membranes. From Fig.1, it appears that to human cells such a nanowire structuring “feels” like a mere surface roughness, while it ‘feels’ like huge obstacles to bacteria.

Reply to major comment 5

We would like to greatly thank for your advices.

(i) As you mentioned, our nanowire structures were not arraigned, but randomly orientated.  We added the following statement to the caption of Figure 1(a): “Although arraigned nanowire structures are simply illustrated here, fabricated nanowires were randomly orientated.”

(ii) The size of bacteria and human cells are generally around 1 and 10-20 µm, respectively.

Comment 6

The nice introduction to biological systems should be completed by a short summary of hydrothermal nanostructuring, especially since bottom-up techniques are generally mentioned (p.3, l.72 and following lines). While it is indeed rare (compared to the hydrothermal synthesis of nanoobjects, such as nanoparticles), hydrothermal nanostructuring of surfaces has been reported for both inorganic materials (https://doi.org/10.1038/s41598-017-04395-0), and organic materials (https://pubs.acs.org/doi/abs/10.1021/acs.macromol.9b00985). I recommend that the authors comment on that.

Reply to major comment 6

We would like to greatly thank the reviewer for suggestions.  We added the statements “Instead of the top-down technologies, bottom-up nanotechnologies allow to synthesize organic (Macromolecules, 2019, 52, 6318-6329) and inorganic (Sci. Rep., 2017, 7, 4155; Chemical Engineering Journal, 2020, 390; Materials Letters, 2012, 67, 8-10; Materials Letters, 2015, 160, 218-221) nanostructures in a self-assembled manner.”, and inserted the references.

Comment 7

Experimental part, p3: (i) spaces between number and unit (300_˚C, etc); (ii) I’d urge the authors to provide more information on the piranha treatment, because the substrates were heated in piranha solution to 180 degC (it could be dangerous for other researchers to reproduce this approach without further instructions, e.g.: which apparatus was used? Flask plus reflux condenser? Autocave?); (iii) p.3, l.100: 300 degC for 3min (which oven was used? How could 300 degC be reached so quickly?); (iv) l. 102 and following lines: In which apparatus was the HT treatment performed?

Reply to major comment 7

We would like to thank your comments.

(i): Thank you very much for your advice.  We inserted spaces between number and unit.

(ii) We added a statement as “The substrate was placed on a petri dish, the piranha solution were poured and the solution was heated at 180 °C on a hotplate for 2 hours.  After cooling, the substrate was took away from the dish by using a Teflon tweezer, washed by Milli-Q water and heated at 300 °C to dry it by using an electric furnace (FB1314M, Thermolyne).”

(iii): For heating at 300 °C, a electric furnace (FB1314M, Thermolyne) was used.  We added a statement as “After cooling, the substrate was took away from the dish by using a Teflon tweezer, washed by Milli-Q water and heated at 300 °C to dry it by using an electric furnace (FB1314M, Thermolyne).”

(iv): The HT treatment was performed by using an oven (DKM400, YAMATO Scientific Co., Ltd.).  We revised as “Each substrate with the ZnO seed layer was immersed into the nanowire growth solution, which was a mixture of 20 mM hexamethylenetetramine (Wako Pure Chemical Industry, Ltd.) and 20 mM zinc nitrate hexahydrate (Alfa Aesar) in Milli-Q water and heated at 95 °C for 5 hours inside an oven (DKM400, YAMATO Scientific Co., Ltd.).”

Comment 8

Characterization: XRD of the nanostructures is necessary. As per SEM, the Zn structure looks like hexagonal wurtzite, but this needs to be proven through XRD. Also, I suppose that the SiO2 layer is amorphous (but this has to be shown through XRD).

Reply to major comment 8

We greatly appreciate the reviewer’s comments.  Previously, we confirmed that hydrothermally synthesized ZnO nanowires were hexagonal wurtzite by XRD technique (J. Phys. Chem. C, 2013, 117, 1197-1203).  However, since SiO2 layer on ZnO nanowires is very thin (~ 10 nm), its XRD signal is expected to be very weak.  Thus, we did not perform the XRD experiment.  However, as you mentioned, XRD-based characterization is very important, in the future work for applying our nanowires to implant application, we will plan to measure XRD spectra of ZnO/SiO2 nanowires on 3-dimenional substrate, which enable to enhance signal intensity.

Comment 9

Motivation: Why were the ZnO nanowires coated with SiO2?

Reply to major comment 9

Since ZnO nanowires can be synthesized relatively mild conditions (95 °C and pH ~7, reference; J. Phys. Chem. C, 2013, 117, 1197-1203) and their structure can easily controlled, we chose ZnO nanowires.  Therefore, we considered that efficient nanostructures (high density of nanowires (> 40 nanowires/µm2), not too small diameter (> 30 nm) and appropriate length (> 200 nm)) for bacteria rupture were easily realized.  However, the structures did not show cell compatibility due to the elution of Zn ions.  The elution can be suppressed via coating the ZnO nanowires with SiO2 layer, thus, we developed ZnO/SiO2 nanowires showing mechanical rupture-based antibacterial and cell-compatibility.

Reviewer 4 Report

The authors reported the synthesis and use of ZnO/SiO2 nanowires for mechanical-rupture based antibacterial activity. This kind of nanostructures has been recently highlighted on the wings of clanger cicadas, and their high antibacterial activity shown to be mechanically-based rupture.

ZnO nanowires are here grown by hydrothermal synthesis from ZnO film, and the coating with SiO2 is done with Atomic Layer Deposition (ALD) techniques. Antibacterial activities are tested with E. Coli, then with HeLa cells to show the selectivity of those nanostructures on bacteria vs human cells.

Even though the mechanical rupture for antibacterial activities is known for few years, and the hydrothermal synthesis of ZnO wires for several years (as for SiO2 ALD), this work highlights the use of hydrothermal synthesis of ZnO nanowires, coated by SiO2 layer with ALD technique, for antibacterial activity via mechanical rupture. Or, as the authors wrote in their conclusion, “This is the first demonstration of a mechanical-based antibacterial and cell compatible nanowires surface fabricated by hydrothermal synthesis” (and, to the best of my knowledge and literature control, this statement is true).

This work is very interesting and deserves to be published in Micromachines after minor corrections.

General comments

  • The authors should not copy/paste the abstract sentences in their introduction part. The sentences highlighted (bold) hereafter fit well in the abstract, but it needed to be developed in the introduction part.

Abstract:

“There are growing interests in mechanical‐based antibacterial surfaces with nanostructures that have little toxicity to cells around the surfaces, however, current surfaces are fabricated via top‐down nanotechnologies, which presents difficulties to apply for bio‐surfaces with hierarchal three‐dimensional structures. Herein, we developed a hydrothermally synthesized nanowire structures having mechanical‐based antibacterial activity and compatibility with human cells. When Escherichia coli were cultured on the surface for 24 hours, over 99% of the bacteria were inactivated, while more than 80% of HeLa cells were cultured on the surface for 24 hours were still alive. This is the first demonstration of mechanical‐based bacterial rupture via the hydrothermally synthesized nanowire structures with antibacterial activity and cell‐compatibility.

Introduction:

“In this work, we developed a bio‐inspired antibacterial surface using nanowires which had a mechanical‐based antibacterial activity and compatibility with human cells (Fig.1b and c). The antibacterial surface had ZnO‐SiO2 core‐shell (ZnO/SiO2) nanowires which were fabricated by hydrothermal synthesis of the ZnO core and followed‐by SiO2 deposition. The hydrothermal synthesis is a promising way to allow the nanowires to be fabricated on large and 3‐dimensional biomaterial surfaces. The ZnO/SiO2 nanowires were 71 nm in diameter, 800‐1,200 nm in height, and approximately 110 nm of average spacing. When bacteria were cultured on the nanowire surface for 24 hours, over 99% of the bacteria were inactivated, and moreover, when HeLa cells as a model of human cells were cultured on the same surface for 24 hours, more than 80% of the cells were still alive and had not ruptured. This is the first demonstration of mechanical‐based bacterial rupture on a cell‐compatible nanowire surface fabricated via hydrothermal synthesis. We expect this antibacterial surface will lead to development of applications for 3‐dimensional biomaterial surfaces.”

  • I strongly suggest to the authors to show more humility within their statements. Too many sentences have personal implications (“we confirmed…; we concluded…; we considered…”) and most of the sentences should be written with a passive form (example: “This statement has been confirmed with these techniques”). Three critical examples that need to be rewritten are listed hereafter (and the list is not exhaustive):
  • Page 9 line 259 (p9 L259) “Therefore, we considered that the present nanowires showed a highly efficient antibacterial property”
  • p10 L285 “This was reasonable because nanostructures have only a small effect on cell viability [34].”
  • p10 L288 “From this result, we judged human cells could be cultured on the ZnO/SiO2

  • The authors claimed “SiO2 film substrate” in the Coli antibacterial test part and “SiO2-deposited ZnO film substrate” with HeLa human cells (cell viability part). Are they two different films? If yes, why the authors did not use the same film? If no, the authors could precise this point in the main text.

  • This point may be related to the previous one, but after showing in the Coli antibacterial test part that the SiO2 film does not show any antibacterial properties, the authors should add a sentence to explain that they won’t use the SiO2 film in the HeLa cells viability part, mainly because of its non-bacterial activity against E. Coli.

  • The authors should add the standard deviation when they measured the average diameter (and spacing) of nanowires (Figure 2a,b). Quick control of the average diameter with SEM images in Figure 2a and ImageJ® software are closed to the ones claimed by the author (20 measures for each image, ZnO nanowires are 59.2 ± 8.6 nm and ZnO/SiO2 nanowires 74.7 ± 17.6 nm) but the average spacing seems overestimated. The authors may explain in the SI how they measured the spacing (central points of the nanowire? From one edge to the closest one?).

  • The authors should add a note when they are talking about Piranha solution. This is a very dangerous oxidant and the general audience should be warned (e.g. to a website link) how to prepare, handle and remove Piranha solution.

Syntax errors

Few points highlighted here:

1/ There is a space between the number and the temperature unit, like 350 °C (e.g. p3 L94 and all along the manuscript);

2/ p4 L31 An antibacterial test: An should be removed when it is a title part

3/ p6L198 strurectures : structures

4/ p10 L300 rapture : rupture

5/ The authors should define once again what is PI (propidium iodide) in the main text (e.g. p9), even if they already defined it in the Part 2. Fluorescence Microscopy observations

Author Response

General comments

The authors reported the synthesis and use of ZnO/SiO2 nanowires for mechanical-rupture based antibacterial activity. This kind of nanostructures has been recently highlighted on the wings of clanger cicadas, and their high antibacterial activity shown to be mechanically-based rupture.

ZnO nanowires are here grown by hydrothermal synthesis from ZnO film, and the coating with SiO2 is done with Atomic Layer Deposition (ALD) techniques. Antibacterial activities are tested with E. Coli, then with HeLa cells to show the selectivity of those nanostructures on bacteria vs human cells.

Even though the mechanical rupture for antibacterial activities is known for few years, and the hydrothermal synthesis of ZnO wires for several years (as for SiO2 ALD), this work highlights the use of hydrothermal synthesis of ZnO nanowires, coated by SiO2 layer with ALD technique, for antibacterial activity via mechanical rupture. Or, as the authors wrote in their conclusion, “This is the first demonstration of a mechanical-based antibacterial and cell compatible nanowires surface fabricated by hydrothermal synthesis” (and, to the best of my knowledge and literature control, this statement is true).

This work is very interesting and deserves to be published in Micromachines after minor corrections.

General comments

The authors should not copy/paste the abstract sentences in their introduction part. The sentences highlighted (bold) hereafter fit well in the abstract, but it needed to be developed in the introduction part.

Abstract:

“There are growing interests in mechanical‐based antibacterial surfaces with nanostructures that have little toxicity to cells around the surfaces, however, current surfaces are fabricated via top‐down nanotechnologies, which presents difficulties to apply for bio‐surfaces with hierarchal three‐dimensional structures. Herein, we developed a hydrothermally synthesized nanowire structures having mechanical‐based antibacterial activity and compatibility with human cells. When Escherichia coli were cultured on the surface for 24 hours, over 99% of the bacteria were inactivated, while more than 80% of HeLa cells were cultured on the surface for 24 hours were still alive. This is the first demonstration of mechanical‐based bacterial rupture via the hydrothermally synthesized nanowire structures with antibacterial activity and cell‐compatibility.”

Introduction:

“In this work, we developed a bio‐inspired antibacterial surface using nanowires which had a mechanical‐based antibacterial activity and compatibility with human cells (Fig.1b and c). The antibacterial surface had ZnO‐SiO2 core‐shell (ZnO/SiO2) nanowires which were fabricated by hydrothermal synthesis of the ZnO core and followed‐by SiO2 deposition. The hydrothermal synthesis is a promising way to allow the nanowires to be fabricated on large and 3‐dimensional biomaterial surfaces. The ZnO/SiO2 nanowires were 71 nm in diameter, 800‐1,200 nm in height, and approximately 110 nm of average spacing. When bacteria were cultured on the nanowire surface for 24 hours, over 99% of the bacteria were inactivated, and moreover, when HeLa cells as a model of human cells were cultured on the same surface for 24 hours, more than 80% of the cells were still alive and had not ruptured. This is the first demonstration of mechanical‐based bacterial rupture on a cell‐compatible nanowire surface fabricated via hydrothermal synthesis. We expect this antibacterial surface will lead to development of applications for 3‐dimensional biomaterial surfaces.”

Reply to general comments

We would like to thank Reviewer 4 for the fairly favorable evaluation of our manuscript.  As the reviewer commented, some revisions are needed for publication in Micromachines.  We acknowledge these important suggestions, and carefully considered them in making the revisions as follows.

We revised the introduction part as the following:

“The ZnO/SiO2 nanowires showed over 99% bacterial inactivation via incubating Escherichia coli for 24 hours.  Moreover, more than 80% of HeLa cells can be cultured on the ZnO/SiO2 nanowires for 24 hours without rupture events.  This is the first demonstration of mechanical-rupture based bacterial rupture on a cell-compatible ZnO/SiO2 nanowire surface fabricated by bottom-up approaches.”

Comment 1

I strongly suggest to the authors to show more humility within their statements. Too many sentences have personal implications (“we confirmed…; we concluded…; we considered…”) and most of the sentences should be written with a passive form (example: “This statement has been confirmed with these techniques”). Three critical examples that need to be rewritten are listed hereafter (and the list is not exhaustive):

Page 9 line 259 (p9 L259) “Therefore, we considered that the present nanowires showed a highly efficient antibacterial property”

p10 L285 “This was reasonable because nanostructures have only a small effect on cell viability [34].”

p10 L288 “From this result, we judged human cells could be cultured on the ZnO/SiO2

Reply to major comment 1

We greatly appreciate the reviewer’s comments.  The statements are revised as the followings:

p9 L259: “Therefore, it was thought that the present nanowires showed a highly effective antibacterial property.”

p10 L285: “Thus, nanostructures have only a small effect on cell viability”

p10 L288: “This result enabled to judge that human cells could be cultured on the ZnO/SiO2 nanowire.”

Comment 2

The authors claimed “SiO2 film substrate” in the Coli antibacterial test part and “SiO2-deposited ZnO film substrate” with HeLa human cells (cell viability part). Are they two different films? If yes, why the authors did not use the same film? If no, the authors could precise this point in the main text.  This point may be related to the previous one, but after showing in the Coli antibacterial test part that the SiO2 film does not show any antibacterial properties, the authors should add a sentence to explain that they won’t use the SiO2 film in the HeLa cells viability part, mainly because of its non-bacterial activity against E. Coli.

Comment 2

We would like to greatly thank for your suggestion.  We added the following statement: “Culturing the cells on the SiO2-deposited substrate was not demonstrated due to showing no antibacterial activity.”

Comment 3

The authors should add the standard deviation when they measured the average diameter (and spacing) of nanowires (Figure 2a,b). Quick control of the average diameter with SEM images in Figure 2a and ImageJ® software are closed to the ones claimed by the author (20 measures for each image, ZnO nanowires are 59.2 ± 8.6 nm and ZnO/SiO2 nanowires 74.7 ± 17.6 nm) but the average spacing seems overestimated. The authors may explain in the SI how they measured the spacing (central points of the nanowire? From one edge to the closest one?).

Reply to major comment 3

We thank for your suggestion.  As for the diameter, we revised the statement as “ZnO nanowires and ZnO/SiO2 nanowires were 51.0 ± 8.2 nm and 71.7 ± 12.1 nm in diameter, respectively, and approximately 110 nm of average spacing of ZnO/SiO2 nanowires was calculated from 83 ± 5 nanowires/µm2 of the surface area density.”  The spacing was calculated with the use of hypothesis that nanowires are vertically orientated and uniformly arraigned.  The calculation is the following:

(Average spacing)=(1/Density of nanowire structures)1/2

=(1/83 (nanowires/um2))1/2

=110 nm

Comment 4

The authors should add a note when they are talking about Piranha solution. This is a very dangerous oxidant and the general audience should be warned (e.g. to a website link) how to prepare, handle and remove Piranha solution.

Reply to major comment 4

We would like to greatly thank for your suggestion.  We added the following statement about using Piranha solution: “The substrate was placed on a petri dish, the piranha solution were poured and the solution was heated at 180 °C on a hotplate for 2 hours.  After cooling, the substrate was took away from the dish by using a Teflon tweezer, washed by Milli-Q water and heated at 300 °C to dry it by using an electric furnace (FB1314M, Thermolyne).”

Comment 5

Syntax errors

Few points highlighted here:

1/ There is a space between the number and the temperature unit, like 350 °C (e.g. p3 L94 and all along the manuscript);

2/ p4 L31 An antibacterial test: An should be removed when it is a title part

3/ p6L198 strurectures : structures

4/ p10 L300 rapture : rupture

5/ The authors should define once again what is PI (propidium iodide) in the main text (e.g. p9), even if they already defined it in the Part 2. Fluorescence Microscopy observations…

Reply to major comment 5

We would like to greatly thank the reviewer for pointing the errors out.  We modified them.

Round 2

Reviewer 2 Report

After implementing Reviewers' suggestions, the quality and clarity of the manuscript have improved. The manuscript can be published in its present form.

Reviewer 3 Report

I think that the manuscript has greatly improved in the revised form. All my comments have been properly addressed and therefor I recommend acceptance of the manuscript in its current form.